# Study on the Changes in the Microcosmic Environment in Forward Osmosis Membranes to Reduce Membrane Resistance

**DOI:** 10.3390/membranes12121203

**Published:** 2022-11-29

**Authors:** Yang Zhao, Liang Duan, Xiang Liu, Yonghui Song

**Affiliations:** 1Chinese Research Academy of Environmental Sciences, Beijing 100012, China; 2State Key Joint Laboratory of Environment Simulation and Pollution Control, School of Environment, Tsinghua University, Beijing 100084, China

**Keywords:** osmotic microbial fuel cell (OsMFC), water flux, membrane resistance, water content, phase transition

## Abstract

Osmotic microbial fuel cells (OsMFCs) are an emerging wastewater treatment technology in bioelectricity generation, organic substrate removal, and wastewater reclamation. To address this issue, proton-conductive sites were strengthened after using the forward osmosis (FO) membrane by reducing the membrane resistance. The mechanism of improving electricity generation was attributed mainly to the unique characteristics of the membrane material and the water flux characteristics of the FO membrane. In particular, only when the concentration of catholyte was greater than 0.3 M was the membrane resistance the main contributor to the overall internal resistance. Meanwhile, through the simulation of the concentration inside the membrane, the changes in the membrane thickness direction and the phase transition of the internal structure of the membrane from the dry state (0% water content) to the expansion state (>50%water content) were analyzed, which were influenced by the water flux, further explaining the important role of the membrane’s microenvironment in reducing the membrane impedance. This further opens a novel avenue for the use of OsMFCs in practical engineering applications.

## 1. Introduction

Microbial fuel cells (MFCs) have attracted wide attention due to their energy advantages, stable operation, and economic advantages, having developed rapidly in recent years [1,2,3]. Utilizing the advantages of both MFCs and FO offers increasingly prominent application potential in wastewater treatment [4,5]. The use of FO membranes can improve the power generation effect in OsMFCs, as shown in Figure 1 [6], as well as the water quality of traditional microbial fuel cells [7,8]; meanwhile, the improvement of the power generation effect of the microbial fuel cell can inhibit the pollution of the FO membrane and the generation of reverse salt flux [9].

OsMFC systems have been extensively studied in recent years, and it has been found ng that they can concentrate the feed solution with power generation and water extraction under a certain osmotic pressure, offering improved economic performance [10]. Fouch’ecour et al. [11] found that the internal resistance of OsMFCs was significantly lower than that of traditional MFCs through the establishment of models and experimental analysis. The model analysis showed that the internal resistance decreased with the increase in the osmotic pressure and water flux. Therefore, OsMFCs can achieve energy recovery and water purification simultaneously in the sewage treatment process. Oh et al. [12] confirmed that OsMFCs promoted ion transport between the anode and cathode compartments due to the generation of water flux, indicating that FO membranes had a lower barrier effect than CEM and AEM membranes as separation materials. Meanwhile, the water flux can also promote proton transfer and alleviate the decrease in anode pH and the increase in cathode pH. These results show that OsMFCs can stabilize the system pH and reduce the system’s overpotential. The solution extracted from OsMFCs usually has a higher salt concentration and lower impedance than that from MFC systems, enabling it to reduce the ohmic loss of the system.

Although the use of FO membranes can affect the membrane impedance in ohmic internal resistance and improve the power generation effect of MFCs, as a characteristic of OsMFCs, the influence of water flux on the power generation capacity of MFCs is worthy of further discussion [13]. At the same time, since the operating conditions of FO membranes present osmotic pressure differences and concentration gradients on both sides of the membrane, insufficient studies have been conducted on the characteristics of the membrane operating under the concentration gradient. In addition, most of the previous studies focused on the comparative analysis of electrochemical indicators, such as electricity generation, while the internal characteristics of the membrane itself—such as the effect of water flux on the impedance of the forward osmosis membrane, along with the distribution of salt concentration in the membrane and its relationship with the membrane impedance—merit in-depth discussion.

In previous studies, the DC method was usually used to measure the membrane impedance and the overall impedance. When current is passing through the membrane, the total impedance of the solution and the membrane can be calculated by measuring the voltage on both sides of the membrane according to Ohm’s law, and then the impedance of the membrane itself can be obtained through the blank group test [14]. However, the ion opposite the surface charge of the membrane is usually used as the charge transfer carrier to transfer the charge through the membrane. In the bulk solution, the charge can be transferred through both cations and anions, so the charge has a difference in the ion transfer number between the membrane and the main solution, leading to the formation of a diffusion boundary layer [15]. At the same time, when the ions transferred to the membrane surface, the double electric layer impedance was also formed on the membrane surface. Both the double electric layer impedance and the diffusion boundary layer hinder the ion transfer. Especially under the condition of low salt concentration, some studies show that the diffusion boundary layer impedance has a large impact. Therefore, the DC method is limited by the test method, and it cannot distinguish the single membrane resistance, diffusion boundary layer impedance, or double electric layer impedance; only the sum of the three impedances can be obtained [16].

In recent years, electrochemical impedance spectroscopy (EIS) measurement technology has gradually matured in the field of electrochemistry. A small disturbance to the system can obtain rich electrode surface information, which is an effective tool for studying electrochemical problems [17]. This approach solves the problem whereby the traditional DC method cannot precisely distinguish between the real membrane electrical resistance and additional impedance. It will also play a key role in further research on the accurate measurement of the membrane electrical resistance and analyses of the mechanisms of high-efficiency production in a forward osmosis microbial fuel cell system [18].

Therefore, in order to investigate the mechanism of the influence of OsMFC membrane impedance on electrical performance, we produced power density curves and polarization curves of OsMFCs and analyzed the internal resistance of MFCs with different membranes. We also investigated the influence of water flux on the impedance of FO membranes, as well as the relationship between OsMFC membrane resistance and external salt solution concentration. Finally, we constructed a membrane resistance model and explored the relationship between the distribution of the salt concentration inside the membrane and membrane resistance.

## 2. Materials and Methods

In this paper, a square double-chambered MFC was used, and FO (polyamide thin-layer composite membrane, TFC membrane, and aquaporin membrane (AQM), HTI), CEM (Neosepta CMX, Tokuyama Soda, Tokyo, Japan), and AEM (Neosepta AMX, Tokuyama Soda) membranes were used as separators. Figure 2a shows the schematic diagram of the OsMFC device, in which the effective volumes of the feed solution and draw solution were 250 mL, and the effective area of the separation membrane was 12.5 cm^2^. Carbon felt (Beijing Sanye Group, Beijing, China) with an effective area of 7.5 cm^2^ was used as the anode material. A platinum-plated carbon paper catalyst (2 mgPt·cm^−2^, Shanghai Hesen Electrode Material Manufacturing Factory, Shanghai, China) was selected as the cathode. An external resistance of 1000 Ω was established between the anode and cathode to record the changes in the voltage of the MFCs. During the experiment, the active layer of the FO membrane faced the feed solution, which was conducive to improving the water flux and reducing the membrane fouling. In order to eliminate the influence of the device, the CEM and AEM separation membranes used the same type of reactor device [19,20,21].

The anodic inoculation of microorganisms was performed from an air-cathode MFC that had been running for half a year. The anolyte used was synthetic wastewater with of the following constituents: NaAc (2 g·L^−1^); NH_4_Cl (0.15 g·L^−1^); NaCl (0.5 g·L^−1^); CaCl_2_ (0.02 g·L^−1^); MgSO_4_ (0.015 g·L^−1^); NaHCO_3_ (0.1 g·L^−1^); KH_2_PO_4_ (0.53 g·L^−1^); K_2_HPO_4_ (1.07 g·L^−1^); and trace elements (1 mL). The cathode chamber used 0.05–1 M NaCl as a catholyte. The initial catholyte was adjusted to neutral pH using NaOH solution. The anolyte and catholyte were connected through an external reservoir, and the peristaltic pump circulated at a flow rate of 20 mL·min^−1^. The water flux was monitored using an analytical balance to calculate the water flux during operation. All experiments were carried out at 25 °C [22].

The output voltage information was collected using a data collector (ZTI, Beijing ZhongHai, China). Under the condition that the MFC voltage remained stable during operation, it was detected and analyzed by the electrochemical workstation (CHI660, Shanghai ChenHua, China). The polarization curve was drawn after the results were processed. The anode was used as the working electrode, while the cathode was used as the counter electrode and reference electrode. The frequency of electrochemical impedance ranged from 100 kHz to 0.01 Hz, and the voltage amplitude was 5 mV. During the experiment, the internal resistance of the three groups of membranes was determined by EIS analysis. The experiment was conducted in an abiotic system composed of the anolyte and catholyte (0.05 M NaCl). The ohmic internal resistance with no membrane was also measured as a blank control [23].

When the concentrations on both sides of the membrane were the same, chronopotentiometry or AC impedance spectroscopy was used to measure the membrane resistance. When the salt concentrations were different, pretreatment experiments were required before the measurements. The membrane was placed in the device shown in Figure 2b, so that the membrane was between two solutions with different concentrations. The concentration on one side was 0.01 M, while that on the other side was 1 M. The solution in each chamber was circulated at a speed of 3 mL·min^−1^ through a peristaltic pump to ensure that the concentration was relatively stable during the experiment. The magnetic stirrer stirred continuously at 800 rpm to fully mix the solution, and the pretreatment time was 2 h [24].

After the pretreatment experiment, the membrane under examination was transferred to the resistance testing device, as shown in Figure 2c, and the orientation of the membrane was consistent with that in the pretreatment device. The voltage was measured by placing Ag/AgCl reference electrodes in a Luggin capillary. The volume of the salt solution was 150 mL, with a circulating flow rate of 15 mL·min^−1^. The purpose of selecting a larger salt solution volume and a smaller membrane area was to ensure the relative stability of the salt solution during the test. Unlike the test with an equal salt concentration, the membrane cannot be removed when the salt concentration is different. In this study, the average impedance value of the low-concentration solution and the high-concentration solution was taken as the overall impedance of the solution, as shown in Equation (1) [24].
(1)$$R_{s}=\frac{R_{high}+R_{low}}{2}$$
where *R_high_* is the impedance of the high-concentration solution, *R_low_* is the impedance of the low-concentration solution, and *R_S_* is the overall impedance of the solution in the presence of a concentration gradient.

The pH of the solution was measured with a pH meter (Mettler Switzerland), and the conductivity of the solution was measured with a desktop conductivity meter (DDSJ-308A, China). To determine the water flux, the balance was used to analyze the change in the weight of the extracted solution over time, and the water flux (L·m^−2^·h^−1^) was determined on the basis of the results, as shown in Equation (2).
(2)$$J_{w}=\frac{V_{p}}{At}$$
where *V_p_* is the change in the draw solution (L) over time *t* (h), and *A* is the effective area of the forward osmosis membrane (m^2^).

## 3. Results and Discussion

### 3.1. Power Generation of MFCs with Different Membranes

Compared with the CEM and AEM membranes, only the FO membrane had the unique aspect of water flux transmission through the water molecules. Figure 3a shows the water flux of the OsMFC under different catholyte concentrations. As there was no osmotic pressure under 0.05 M NaCl, no water flux was produced. When the concentration of the catholyte increased to 1 M NaCl, the water flux of the OsMFC began to increase gradually, but it decreased significantly after 24 h due to the dilution of the draw solution and the adverse effects of membrane pollution factors. Meanwhile, the catholyte’s conductivity changed significantly, from 96 to 52.3 ms·cm^−1^, indicating that the catholyte was diluted, which reduced the osmotic pressure difference and had a negative impact on the water flux [25]

Figure 3b shows the voltage changes of each type of MFC membrane. Under the condition of a 1 M NaCl catholyte, the OsMFC system’s voltage was 0.65 V—significantly higher than that of the other MFC membranes. After four cycles of operation, the output voltage of the OsMFC decreased, which may have been related to the gradual increase in the reverse salt flux. A large amount of salt solution migrated and accumulated on the side of the feed solution, inhibiting the microbial activity [26]. In general, using an FO membrane as an MFC separation material can achieve a better power generation effect than an ion-exchange membrane [27].

### 3.2. Comparison of Cathode and Anode Potential

In the operation cycle, a 35 g·L^−1^ NaCl solution was used as the catholyte to compare the open-circuit voltage of the OsMFC and MFC. As shown in Figure 4a, the open-circuit voltage of both the OsMFC and MFC was equal to 0.76 V. After 6 h, the open-circuit voltage of the OsMFC started to drop rapidly, while the open-circuit voltage of the MFC started to drop after 9 h, indicating that the increase in the anode salt concentration led to an increase in the anode potential, thereby reducing the open-circuit voltage [28]. At the same time, the potential of the cathode and anode was measured. The results showed that the cathode potential of the OsMFC and MFC remained relatively stable within the observed period, and the cathode potential of the MFC remained at ~0.3 V. The cathode potential of the OsMFC was slightly lower than that of the MFC, remaining at ~0.28 V, which may have been due to the water flux reducing the cathode conductivity and the cathode potential of the OsMFC. The anode potential of the two MFCs remained at −0.5 V at the beginning and rose to −0.2 V at 6 h, and then it rose again to −0.3 V after 9 h. This explains why the open-circuit voltage of the MFC decreased after 9 h—due to the increasing anode potential. As for the water flux of the OsMFC, the volume of the anolyte per operation cycle decreased from 250 mL to 105 mL, which caused the concentration of the anolyte to be higher than in the MFC. At the same time, there were pH differences between the two types of MFC. After an operation cycle, the pH of the OsMFC’s catholyte was 8.5, while that of the MFC was 10.3. The difference between the two pH values may be due to the fact that the proton transfer of the FO membrane was better than that of the CEM membrane, so the OsMFC could reduce the pH through the addition of more H+ to the cathode chamber [29]. On the other hand, the unique water flux characteristics of the FO membrane diluted the OH—produced by cathodic reduction [30].

As shown in Figure 4b, under the closed-circuit conditions, during an operation cycle, the current density of the OsMFC was 100 A·m^−2^, while that of the MFC was 50 A·m^−2^, which decreased to less than 40 A·m^−2^ when operating for 8 h. At the same time, the cathode and anode potentials were also measured. It can be seen that the cathode and anode potentials of the OsMFC and MFC were the same: the anode potential was −0.42 V and the cathode potential was 0.2 V. This shows that the high current density of the OsMFC was not caused by its cathode and anode potentials [31]. Therefore, according to the analysis of the composition of the internal resistance of the fuel cells, it was proposed that the rising resistance was caused by the separation materials of the two fuel cell systems with different membrane impedance [32].

### 3.3. Internal Resistance of MFCs with Different Membranes

As shown in Figure 5a, the internal resistances of the MFC systems composed of different membranes (i.e., FO, CEM, and AEM) were further determined by polarization curves. The maximum power of the OsMFC was 7.3 W·m^−3^ with the FO membrane, followed by 5.3 W·m^−3^ for the AEM and 5.6 W·m^−3^ for the PEM. The internal resistance of the OsMFC under the FO membrane was 70 Ω, which was significantly lower than that of the CEM (100 Ω) and AEM (103 Ω), according to the slope of the polarization curve. This proved that the power generation performance can be improved by using an FO membrane [33], indicating that the OsMFC can greatly improve power generation because of its relatively lower impedance [34], but the impedance obtained at this time was not only membrane impedance—it also included double electric layer impedance and diffusion boundary layer impedance [35].

### 3.4. Effect of Water Flux on Membrane Impedance

The results showed that the use of the FO membrane improved the power density of the MFC and reduced the overall internal resistance, while the difference between the FO and the ion-exchange membrane feed was the former’s unique water flux characteristics [36]. Since the electro-osmosis and osmotic pressure could promote the forward transmission of water flux in the OsMFC, more water could pass through the membrane when the osmotic pressure difference and current density increased. As shown in Figure 6a, when the external resistance was 117.6 Ω and the osmotic pressure was 43 atm, the current density, water flux, and membrane impedance were 1.8 A·m^−2^, 5.3 LMH, and 25 Ω·cm^2^, respectively [37].

Meanwhile, under the open-circuit voltage, the water flux was 5 LMH and the membrane resistance was 89 Ω cm^2^. To further analyze the relationship between the water transfer amount and the membrane resistance due to the electric drag, the osmotic pressure on both sides of the membrane remained unchanged when the concentration of the anode and cathode was unchanged. When the external resistance changed from 2358 Ω to 117.6 Ω, the current density increased from 0.05 Am^−2^ to 1.8 A m^−2^, while the membrane impedance reduced from 89 Ω cm^2^ to 25 Ω cm^2^.

Since osmotic pressure produced water flux and affected the electricity generation, the membrane impedance was measured under different osmotic pressure. In order to eliminate the influence of electric drag on the water flux, experimental groups with different osmotic pressure under the same current were constructed with four test currents and water flux conditions, as shown in Figure 6b. It can be seen that the maximum membrane resistance was 89 Ω·cm^2^ under open-circuit voltage, with 43 atm osmotic pressure and 5 LMH corresponding water flux [38].

When the osmotic pressure increased to 106 atm, the water flux rose to 10.3 LMH, while the membrane impedance reduced to 51 Ω cm^2^. When increasing the osmotic pressure to 137 atm, the water flux rose to 11.5 LMH, and the membrane resistance reduced to 7.3 Ω cm^2^. Therefore, increasing the current and osmotic pressure can reduce the membrane impedance by influencing the water flux.

### 3.5. Comparison and Analysis of Membrane Impedance

It can be seen from Table 1 that the use of an FO membrane can effectively improve the power generation of the MFC. According to the correlation of membrane resistance and the low-concentration side of the membrane, the key factor affecting the power generation was the low-concentration side, because the low concentration brought higher solution self-impedance and higher membrane impedance, which increased the overall internal resistance. The experiment showed that when the concentration of the catholyte increased from 2 to 35 g·L^−1^, the water flux of the OsMFC increased from 0 to 3.58 LMH, and the current increased from 42 to 65 mA, which led to the overall impedance decreasing from 31.36 to 8.16 Ω.

As a comparison, the overall impedance of the MFC with a CEM decreased from 35.23 to 11.67 Ω, whereas the impedance of the OsMFC catholyte decreased from 12.17 to 0.45 Ω, and the impedance of the MFC catholyte decreased from 11.15 to 0.31 Ω. The reason for this may be the OsMFC’s water flux from anode to cathode, which diluted the catholyte concentration and caused the loss of reverse flux from cathode to anode [39]. The increase in the catholyte’s concentration also affected the FO membrane’s impedance. When the concentration increased from 2 to 35 g·L^−1^, the membrane impedance decreased from 17.6 to 5 Ω, while the CEM membrane impedance decreased from 22.6 to 9.63 Ω.

By comparing the internal resistance composition of the OsMFC and MFC, it can be seen that when the concentration of anolyte is kept constant, the concentration of the catholyte can significantly affect the membrane resistance and electrolyte impedance, thereby affecting the overall internal resistance. The lower the concentration, the greater the impedance of each part, leading to greater overall internal resistance. Similarly, when the concentration of the catholyte increased, the impedance of each part decreased. Moreover, the decrease in the catholyte’s impedance was faster than that of the membrane’s impedance. When the concentration of the catholyte increased, no matter what type of membrane was used, the contribution to the overall impedance came mainly from the impedance of the membrane itself. When the concentration of the catholyte increased from 2 g·L^−1^ to 35 g·L^−1^, the proportion of the OsMFC membrane impedance in the overall internal resistance increased from 56% to 61.2%, while the proportion of the MFC membrane impedance in the overall internal resistance increased from 64.1% to 82.5%. Compared with the CEM membrane, the FO membrane contributed a smaller proportion of the overall internal resistance due to its own low membrane impedance.

The proportions of each component in the overall internal resistance were analyzed. For the CEM membrane, when the anolyte was less than 17 g·L^−1^, i.e., 0.29 M, the overall internal resistance decreased rapidly with the increase in the concentration, and the internal resistance was due mainly to solution impedance and membrane impedance. When the concentration of the anolyte was more than 17 g·L^−1^, the membrane impedance contributed most of the internal resistance, and the membrane impedance essentially did not change with the concentration of the solution. Meanwhile, for the FO membrane, the critical solution concentration was 23 g·L^−1^, i.e., 0.39 M, and when the catholyte was less than 0.39 M, the membrane impedance and solution impedance were the main contributors to the overall internal resistance, while when the catholyte concentration was greater than 0.39 M, the membrane resistance became the main contributor, as shown in Figure 7.

### 3.6. Measurement of the Membrane Resistance and Electrolyte Resistance of the Forward Osmosis Membrane

We found that the water flux could be directly influenced by the osmotic pressure on both sides of the membrane, along with the current density, thereby influencing the membrane impedance. In order to obtain more detailed information about the membrane impedance, the EIS test method can be used in an abiotic system. In order to verify that the concentration of the device shown in Figure 4 remained relatively stable during the measurement, the chronopotentiometric method was used to scan in the direction of current increase for 15 min (exceeding the measurement time) to check whether the solution’s concentration changed during the membrane impedance measurement [40].

The results shown in Figure 8a indicate that the variation range of membrane impedance was kept within 5%, so it could be considered that the concentration of solution on both sides of the membrane did not change during the membrane resistance test.

#### 3.6.1. Membrane Impedance and Impedance Model at the Same Concentration

The impedance of the FO membrane, double electric layer, and diffusion boundary layer was measured from the change in the circulating velocity by setting an equivalent circuit and using AC impedance spectroscopy. When the catholyte had a low salt solution concentration (2 g·L^−1^ NaCl solution) and the osmotic pressure between the catholyte and anolyte was equal, no water flux was generated during the operation. Figure 8b shows the membrane impedance measured by DC and AC methods when the concentration on both sides of the membrane was the same [41]. It can be seen that when the solution concentration was low—i.e., less than 0.2 M—the impedance value measured by the DC method was greater than that measured by the AC method. When the solution concentration was 0.2 M, the membrane resistance values measured by the two methods were generally close. Therefore, under the condition of low salt concentration, the real impedance value cannot be measured by the DC method when measuring the membrane impedance because, in this case, the main impedance affecting the system was the solution impedance, while the membrane impedance remained at a low level.

Table 2 compares the overall resistance values obtained by the aforementioned methods; it can be seen that the total resistance values obtained by the DC and AC methods were close. The diffusion boundary layer resistance accounted for more than 76% of the total resistance, illustrating that increasing the circulating velocity of the solution can reduce the resistance when using the DC method. When the salt concentration was high, the resistance of the membrane itself played a major part, and the results were consistent with the inference obtained by the DC test method. The resistance of the diffusion boundary layer still had a high contribution ratio when the concentration was high.

Assuming that the solution inside the membrane affecting the membrane impedance was divided into conductive and non-conductive regions, the membrane impedance can be expressed as follows [42]:(3)$$R_{m}=a+\frac{b}{c}$$
where a and b are the impedance of the non-conductive area and the impedance of the conductive area, respectively, while c is the concentration of the internal salt solution. When the concentration of the solutions on both sides of the membrane was the same, the concentration of the conductive region was the same as that of the external salt solution, as shown in Figure 8c.

The results showed high correlation. The resistance of the non-conductive area was y0 = 18.6, while the impedance of the conductive part was negatively correlated with the concentration of the external solution, and the correlation coefficient (R^2^) was 0.988.

#### 3.6.2. Membrane Impedance at Different Concentrations

As the FO membrane is often operated in a salinity gradient environment, it was necessary to investigate the membrane resistance under the conditions of concentration differences. When the concentration of the feed solution was kept at 0.1 M and the concentration of the draw solution was increased, the membrane resistance still remained in a high range, only decreasing from 203.37 Ω cm^2^ to 165.38 Ω cm^2^, as shown in Figure 9. This result indicated that the membrane resistance was affected mainly by the side with the lower salt concentration. After increasing the concentration at the lower side from 0.1 M to 0.3 M, the membrane resistance reduced to below 21.07 Ω cm^2^, and the minimum membrane resistance was 18.30 Ω cm^2^, which was obtained when the concentration of the draw solution was 1 M NaCl. Therefore, increasing the concentration of the salt solution on the lower side can effectively reduce the membrane resistance. In addition, when the lower-concentration side was increased to 0.3 M, it was very close to the minimum value of membrane impedance, while further increases in the concentration had little impact on the membrane impedance. Therefore, there was a limit to how far the membrane resistance could be reduced by increasing the salt concentration. When the concentration increased to the limiting concentration, the reduction in membrane impedance gradually stabilized in the range near the limiting value [43].

Similarly, the chronopotentiometric method was the most common method used to measure membrane and solution resistance (R_s+m_) within the same concentration gradient variation range. As shown in Table 3, when the concentration of the salt solution on one side of the membrane was kept at 0.1 M while the other side increased, the membrane impedance decreased from 719.26 Ω cm^2^ to 192.95 Ω cm^2^. When the low-concentration side increased to 0.3 M, R_s+m_ was only 26.19 Ω cm^2^, indicating that R_s+m_ and individual R_m_ were affected by the low-concentration side of the salt solution. Therefore, the sum of membrane impedance and solution impedance obtained by chronopotentiometry (R_s+m_) and the membrane impedance (R_m_) obtained by AC impedance spectroscopy (EIS) decreased with the increase in the concentration on the low-concentration side, and when the concentration reached 0.3 M, the reduction rates of R_s+m_ and R_m_ tended to be gentle and in a steady state [44].

When there was a concentration difference between the solutions on either side of the membrane—i.e., c1 ≠ c2—as shown in Table 4, the solution concentration within the membrane was involved. According to the negative correlation between the membrane resistance of the FO membrane and the salt solution concentration on the low-concentration side, the concentration of the conductive part of the membrane was equivalent to that of the external membrane’s lower concentration; that is, there was a relationship of ci≈c_low_, but the concentration of the solution in the membrane near the side with the higher concentration was similar to that of the side with the higher concentration, due to the osmotic pressure and salt concentration gradient, c_i_≈c_high_. The calculation of the solution concentration of the conductive part of the membrane was as follows [45]:C_i_ = C_low_ + (C_high_ − C_low_)(x/δ)^n^(4)
where x is the position (m) at different thicknesses inside the membrane, δ is the membrane thickness (m), and n is used to reflect the concentration difference between two points in the membrane under a certain salt solution concentration difference. In particular, when n = 1, the concentration of the solution inside the membrane changed linearly; the specific change trend is shown in Figure 10a.

Figure 10a shows the changes in the solution concentration in the membrane with membrane thickness under different salt solution concentration gradients when water moved from the low-salt-concentration side to the high-salt-concentration side as a result of the osmotic pressure. When the solution concentration on both sides of the membrane remained unchanged, the greater the water flux, the faster the rate of the solution increased in the membrane, leading to a smaller n value. When the concentration on the low-concentration side increased to the limiting value, the membrane resistance was determined by the non-conductive area, so the concentration of the solution in the conductive area within the membrane changed linearly with n = 1. According to the different values assigned to the coefficient n, the solution concentrations at different positions in the membrane under a fixed concentration difference were obtained, and the reasonable coefficient (n) of the test membrane under this concentration gradient was finally determined on the basis of the difference from the real membrane resistance [46].

As shown in Table 5, by comparing the calculations with the real membrane impedance, we found when N = 1.93, the difference between the calculated result and the real membrane impedance was the smallest. The membrane impedance measures at different thicknesses in the membrane were then obtained, as shown in Figure 10b.

As shown in Table 6 and Table 7, the variation trend of membrane impedance still increased with the decrease in the concentration of the solution on both sides of the membrane. When the salt concentration was above 0.1 M, the difference between the simulated impedance and the measured impedance was small, and the difference was kept below 2 Ω·cm^2^. When the concentration on the low-salt-concentration side was 0.01 M, the simulated impedance of the model deviated greatly from the experimental impedance, with a maximum deviation of 11.93. This shows that the real membrane resistance was significantly affected by the solution impedance at low salt solution concentrations, so the predicted result was greater than the simulated impedance value.

### 3.7. Internal Impedance Combination Mode

Since the conductivity of membrane ions was influenced mainly by the counterions and fixed charges on the membrane surface, there was an extremum where the counterions and fixed charges on the membrane surface were affected by the concentration outside the membrane [47]. The results presented in Figure 10a show that there was a correlation whereby the membrane resistance decreased with the increase in the concentration when the salt solution was less than 0.3 M. This phenomenon can be attributed to the structure of the membrane itself. Assuming that the membrane was a homogeneous system, as the concentration of the external solution was greater than 0.3 M, the membrane impedance should have continued to decrease. However, the results shown in Figure 10b reveal that the membrane resistance was fixed within a certain range. The conductivity of the membrane was closely related to the movable ions in the solution inside the membrane, and the solution inside the membrane could be divided into two parts: one was the non-conductive part, which was not affected by the concentration of the external solution, and the other part was the conductive part, which was affected by the concentration of the external solution. If the connection mode of the two parts was in parallel, this meant that the overall membrane conductivity would increase linearly when the conductivity of the conductive part increased [48], which was clearly contrary to the previous conclusion. Therefore, with the increase in the concentration of the external solution, the membrane conductivity first increased and then tended to stabilize in a certain range.

Conversely, if the solution inside the membrane was in series, the membrane conductivity changed nonlinearly when the conductivity of the conductive part increased, but there was a limit value that was consistent with the experimental results. Therefore, the solution inside the membrane affecting the membrane impedance could be divided into two parts: one part related to the external salt solution concentration, and the other part unrelated to the external salt solution concentration. The connection of the two parts was a combination of series connections.

### 3.8. Water Content and Phase Transition in the Membrane

Like the ion-exchange membrane, proton-conductive sites were attached to the FO membrane by employing the -SO_3_H group to improve the proton transfer rate. These hydrophilic groups absorbed environmental water molecules to form hydrophilic clusters, as shown in Figure 11. When the membrane water content increased, these hydrophilic clusters would grow and form an osmotic pressure system within the membrane [49]. However, the water content was relatively too low to form a conductive area or an ionic phase area. With the increase in the water content, these hydrophilic clusters began to connect with one another through the ion permeation channel under the effect of the permeation system. When the water content increased to 50%, the phase inside the membrane changed, and the non-conductive phase state converted to the conductive ion phase state, thereby influencing the membrane impedance. Therefore, the changes in the phase state of the membrane with the changes in the water content and membrane internal osmotic system caused by the water flux were the fundamental reasons for the change in membrane resistance [50].

## 4. Conclusions

In this study, the OsMFC constructed by coupling an FO membrane with an MFC was able to obtain higher power density and lower overall internal resistance. The mechanism of improving electricity generation was attributed mainly to the unique membrane material characteristics and water flux characteristics of FO membrane.

By analyzing the relationship between the concentration of the catholyte and the membrane resistance of the FO and CEM membranes, we found that the limiting values (k) of the catholyte concentration were 0.29 M and 0.4 M, respectively. Only when the concentration of the catholyte was greater than k was the membrane resistance the main contributor to the overall internal resistance.

In order to verify the limiting value k, the membrane impedance of different concentrations was further analyzed in an abiotic system, and it is believed that the key factors to determine the membrane impedance were the ionic phase part inside the membrane and the corresponding concentration of this part—that is, the concentration of the solution inside the membrane was the key factor in determining the membrane impedance.

Through the model simulation of the concentration inside the membrane, the changes in the membrane’s thickness direction and the phase transition of the internal structure of the membrane from the dry state to the expansion state were analyzed, and it was found that they were influenced by the water flux, further explaining the important role of the membrane’s internal microenvironment in reducing the membrane impedance.

## Figures and Tables

**Figure 1 membranes-12-01203-f001:**
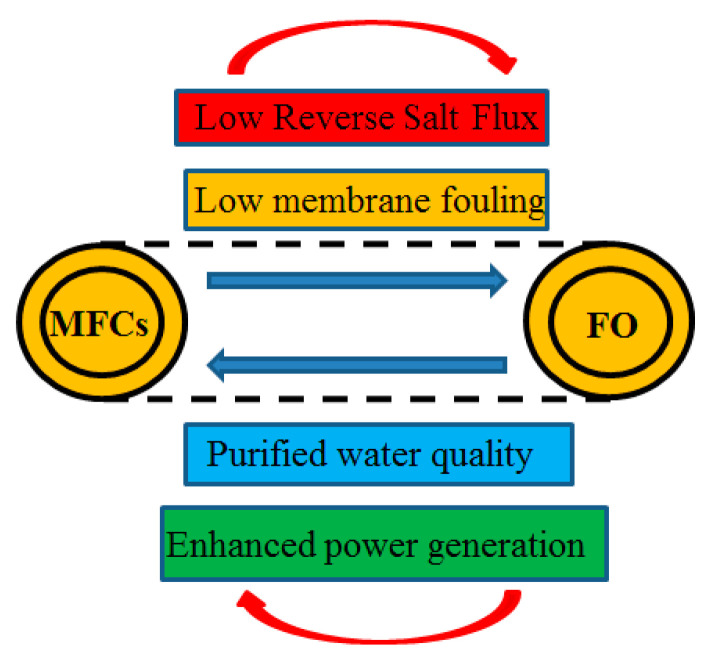
Coupling advantages of forward osmosis technology and microbial fuel cells.

**Figure 2 membranes-12-01203-f002:**
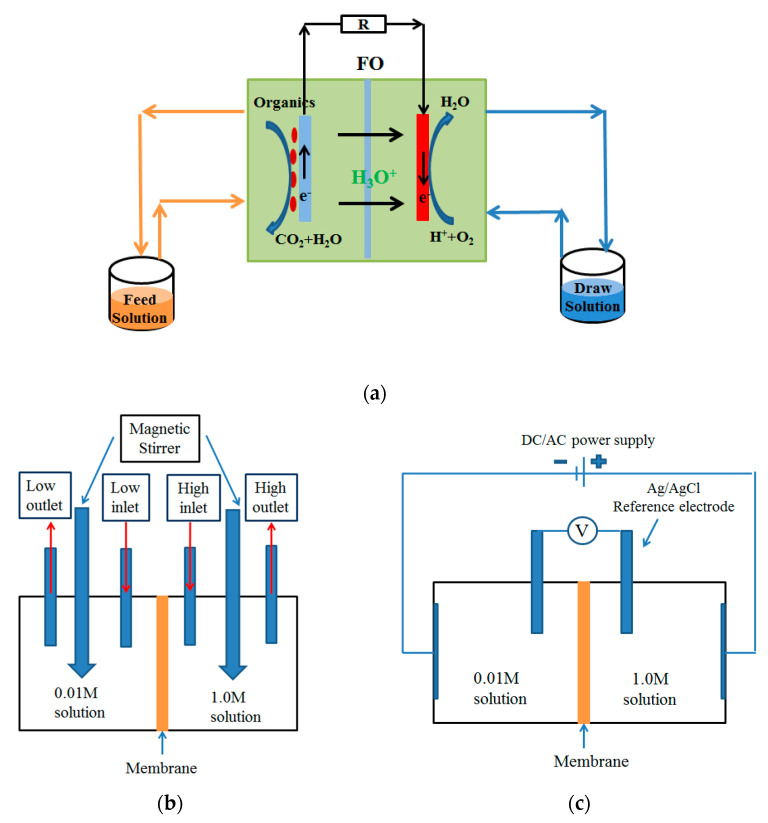
(**a**) Schematic of the forward osmosis microbial fuel cell (OsMFC). (**b**) Pretreatment device for measuring membrane resistance under the concentration gradient. (**c**) Membrane impedance measuring device with different concentrations on both sides of the membrane.

**Figure 3 membranes-12-01203-f003:**
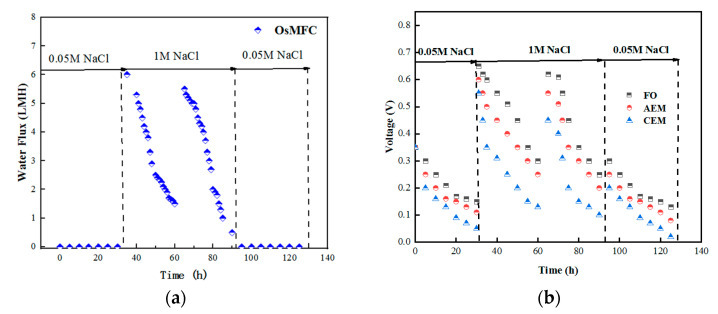
(**a**) Variation of water flux in the OsMFC system; (**b**) power generation trends of MFC systems with different membrane compositions.

**Figure 4 membranes-12-01203-f004:**
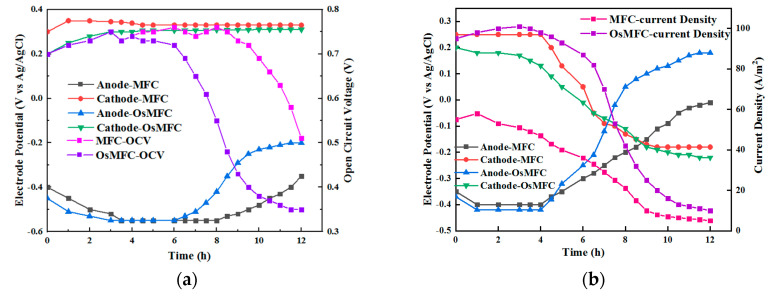
(**a**) Comparison of current density and cathode and anode potentials under open-circuit conditions; (**b**) comparison of current density and cathode and anode potentials under closed-circuit conditions.

**Figure 5 membranes-12-01203-f005:**
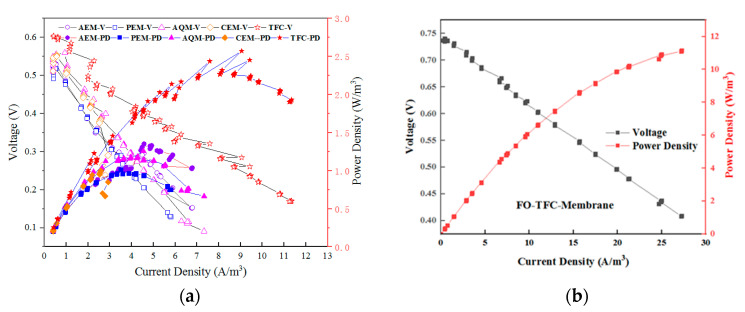
(**a**) Polarization curves and power density curves of the different partition materials; (**b**) changes in polarization curves after increasing the concentration of the draw solution/catholyte.

**Figure 6 membranes-12-01203-f006:**
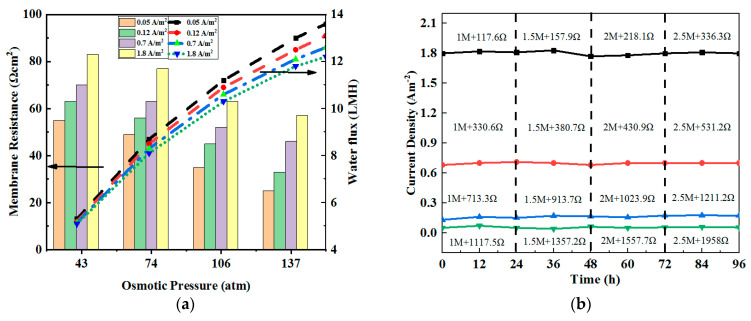
(**a**) Interaction between membrane impedance and current/water flux; (**b**) similar currents obtained under different conditions (catholyte concentration of 1 M NaCl + 117.6 Ω external resistance; ~2.5 M NaCl catholyte concentration + 2358 Ω external resistance).

**Figure 7 membranes-12-01203-f007:**
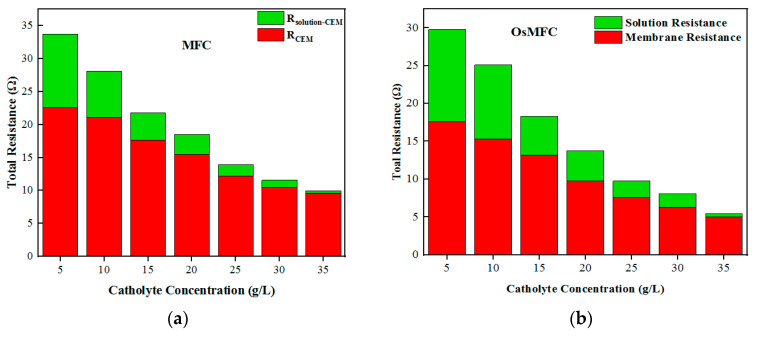
Changes in membrane impedance and solution impedance with solution concentration.

**Figure 8 membranes-12-01203-f008:**
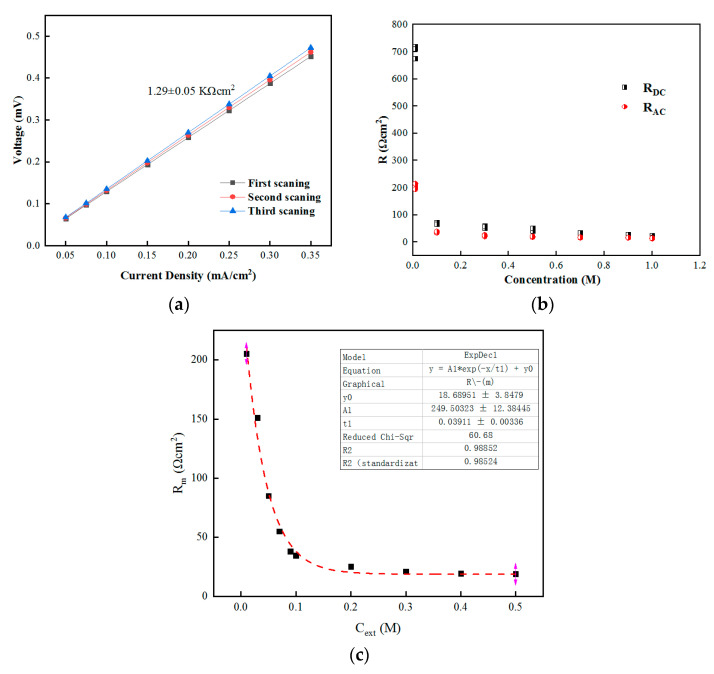
(**a**) Membrane impedance at different NaCl concentrations on either side of the membrane: 0.01 M and 1 M, respectively; (**b**) membrane impedance measured by DC and AC methods at the same concentration on both sides of the membrane. (**c**) Correlation analysis between the experimental membrane impedance and model-calculated membrane impedance.

**Figure 9 membranes-12-01203-f009:**
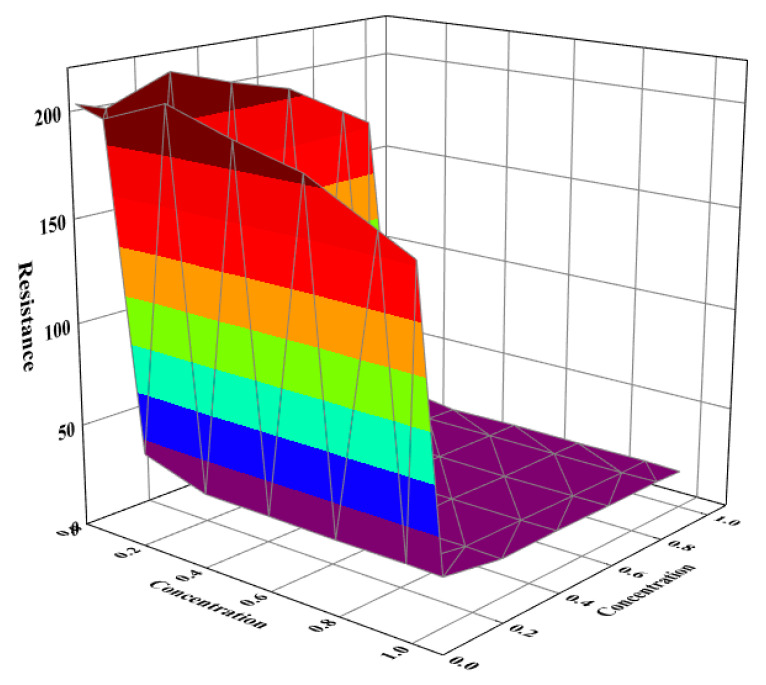
Membrane resistance of different concentrations on either side of the membrane.

**Figure 10 membranes-12-01203-f010:**
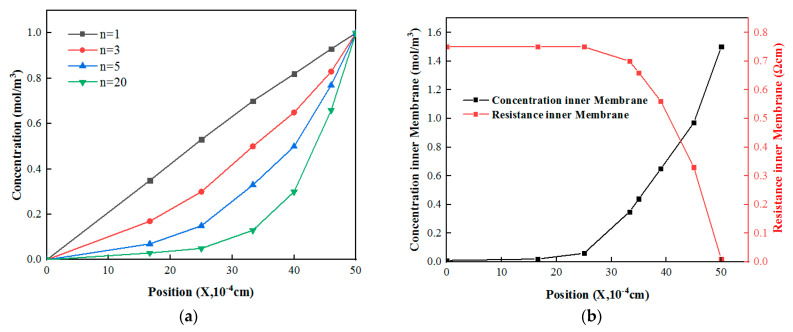
(**a**) Ion concentration changes at different positions in the membrane under different n values; (**b**) solution concentration and membrane resistance at different thicknesses in the membrane.

**Figure 11 membranes-12-01203-f011:**
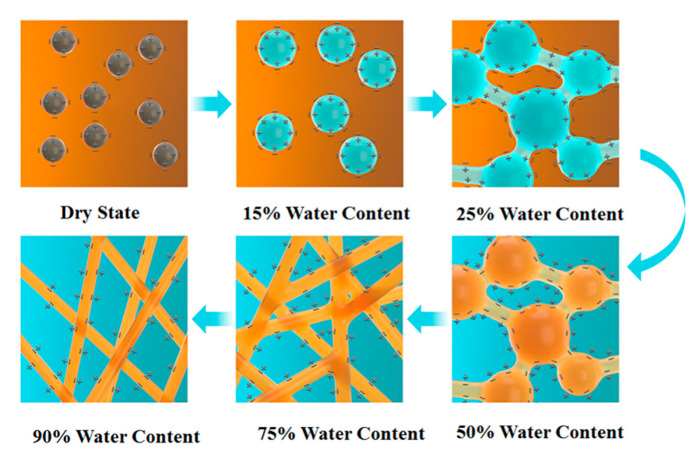
Changes in the phase state in the membrane with water content.

**Table 1 membranes-12-01203-t001:** Comparative analysis of internal resistance composition between the OsMFC and MFC.

Resistance (Ω)	OsMFC (TFC)			MFC (CEM)		
Catholyte (g/L)	Membrane Resistance	Solution Resistance	Total Resistance	Membrane Resistance	Solution Resistance	Total Resistance
5	17.6	12.17	31.36	22.6	11.15	35.23
10	15.3	9.78	28.31	21.1	7.02	31.15
15	13.2	5.12	21.32	17.6	4.19	23.73
20	9.8	3.95	16.71	15.5	3.03	19.33
25	7.6	2.13	11.38	12.2	1.76	15.27
30	6.3	1.78	9.15	10.5	1.05	12.66
35	5	0.45	8.16	9.63	0.31	11.67

**Table 2 membranes-12-01203-t002:** Comparison of impedance values measured by DC and AC impedance spectroscopy.

Type	Flow Rate (mL·min^−1^)	0.05 M (Ω cm^2^)					0.5 M (Ω cm^2^)				
		*R_DC_*	*R_Ac_*	*R_DL_*	*R_DBL_*	*R_M_*	*R_DC_*	*R_Ac_*	*R_DL_*	*R_DBL_*	*R_M_*
CEM	100	460	386	93	337	23	24	22.6	2	7.6	13
	500	220	201	28	145	18	16	15.85	0.75	3.5	11.6
	800	130	128	25	80	13	13.6	13.4	0.3	2.1	11
AEM	100	400	255	65	300	20	13.3	12.6	1.1	6	5.5
	500	150	98	21	100	17	9.5	8.4	0.9	2.6	5
	800	80	68	3	55	10	8.3	7.75	0.85	2	4.9
FO	100	230	147	17	125	5	10.85	10.05	0.25	6.5	3.3
	500	100	87	11	70	6	6.52	6.08	0.23	2.75	3.1
	800	40	35	9	21	5	5.8	5.22	0.22	2	3

**Table 3 membranes-12-01203-t003:** The membrane impedance R_s+m_ (Ω cm^2^) at electrolytes C1 and C2 (M) on both sides of the membrane, as measured by DC chronopotentiometry.

C1\C2	0.01	0.1	0.3	0.5	0.7	0.9	1.0
0.01	719.26	509.99	444.91	277.18	271.28	253.94	192.95
0.1	509.98	70.14	52.98	51.05	50.81	51.11	46.42
0.3	444.90	52.98	26.19	27.21	26.31	26.13	24.86
0.5	277.18	51.05	27.21	20.65	23.66	23.54	23.12
0.7	271.28	50.81	26.31	23.66	19.51	22.22	21.25
0.9	253.94	51.11	26.13	23.54	22.22	20.89	21.19
1.0	192.95	46.42	24.86	23.12	21.25	21.19	20.53

**Table 4 membranes-12-01203-t004:** The membrane impedance R_m_ (Ω cm^2^) at electrolytes C1 and C2 (M) on both sides of the membrane, as measured by AC impedance spectroscopy.

C1\C2	0.01	0.1	0.3	0.5	0.7	0.9	1.0
0.01	203.37	198.85	210.77	200.30	191.75	174.17	165.38
0.1	198.85	34.56	24.56	24.20	23.60	23.60	23.30
0.3	210.77	24.56	21.07	19.99	18.42	18.76	18.40
0.5	200.30	24.20	19.99	18.81	18.76	18.76	18.96
0.7	191.75	23.60	18.42	18.76	18.33	18.60	18.72
0.9	174.17	23.60	18.76	18.76	18.60	18.57	18.88
1.0	165.38	23.30	18.40	18.96	18.72	18.88	18.30

**Table 5 membranes-12-01203-t005:** Concentration of solution in the membrane, impedance in the membrane, and difference from the real impedance under different conditions.

N	Position Inside Membrane	Concentration Inside Membrane (mol·m^−3^)	Resistance Inside Membrane (Ω·cm)	ΔN Membrane Resistance Difference
1	1/3	0.40	17.35	3.718
	1/2	0.55	16.26	1.756
2	1/3	0.20	21.34	−0.272
	1/2	0.33	18.27	−0.253
3	1/3	0.13	25.33	−4.262
	1/2	0.21	20.87	−2.853
1.9	1/3	0.21	20.90	0.166
	1/2	0.34	18.04	−0.020
1.95	1/3	0.21	21.12	−0.053
	1/2	0.33	18.16	−0.136
1.93	1/3	0.21	21.04	0.035
	1/2	0.34	18.11	−0.049
1.92	1/3	0.21	20.99	0.078
	1/2	0.34	18.09	−0.066

**Table 6 membranes-12-01203-t006:** R_M- model_ (Ω cm^2^) of the relationship between the calculated resistance and the concentration of the external salt solution.

C1\C2	0.01	0.1	0.3	0.5	0.7	0.9	1.0
0.01	203.30	191.62	186.98	184.87	183.49	182.47	181.68
0.1	191.62	34.80	27.33	24.74	23.30	22.39	21.67
0.3	186.98	27.33	22.33	20.83	20.05	19.50	19.08
0.5	184.87	24.74	20.83	19.87	19.20	18.84	18.54
0.7	183.49	23.30	20.05	19.20	18.78	18.42	18.18
0.9	182.47	22.39	19.50	18.84	18.42	18.18	17.94
1.0	181.68	21.67	19.08	18.54	18.18	17.94	17.82

**Table 7 membranes-12-01203-t007:** The deviation between the membrane impedance R_m_ and the model resistance R_m model_ (Ω cm^2^).

C1\C2	0.01	0.1	0.3	0.5	0.7	0.9	1.0
0.01	0.07	7.23	9.79	11.93	8.26	−8.3	−9.3
0.1	7.23	−0.24	−1.77	−0.54	0.3	1.21	1.63
0.3	9.79	−1.77	−1.26	−0.84	−1.63	−0.74	−0.68
0.5	11.93	−0.54	−0.84	−1.06	−0.44	−0.08	0.42
0.7	8.26	0.3	−1.63	−0.44	−0.45	0.18	0.54
0.9	−8.3	1.21	−0.74	−0.08	0.18	0.39	0.94
1.0	−9.3	1.63	−0.68	0.42	0.54	0.94	0.48

## Data Availability

The data presented in this study are available upon request from the corresponding author.

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
