# Peer review of "Study on the Changes in the Microcosmic Environment in Forward Osmosis Membranes to Reduce Membrane Resistance"

_membranes, 2022, doi:10.3390/membranes12121203_

Round 1

Reviewer 1 Report

Title: -Study on the Change of Microcosmic Environment in Forward Osmosis Membrane to Reduce Membrane Resistance.

Manuscript ID: Membranes-2033968

In general, this is a solid work generated some interesting and reliable results. The author focused on forward osmosis membrane in Osmotic microbial fuel cell to improve the performance of bioelectricity generation, organic substrate removal, and wastewater reclamation by reducing membrane resistance. However, the report of this work has some significant drawbacks, and some claims are even misleading, I suggest accepting this work after major correction. In addition, some suggestions basing on the insufficient justification and analysis of data in this present work are also given, as shown in follows:

1.       Given the topic and scope of the paper, some recently MFC work (Frontiers in Nanotechnology, 4 (2022) 1–16; Chemical Engineering Journal Advances, 10 (2022) 100283; Sensing and Bio-Sensing Research, 36 (2022) 100484; Journal of Chemical Reviews, 2021 3(4) 320–344; Journal of Nanomaterials, 2021 (2021) 1–21; Nano-Structures & Nano-Objects, 25 (2021) 100663; All Life, 14 (2021) 541–568; Materials Science for Energy Technologies, 3 (2020) 136–149; Journal of Science: Advanced Materials and Devices, 4 (2019) 353–369) should be highlighted in the introduction and discussion part to broaden the readership.

2.       Please carefully check the sentences again. I strongly encourage the authors to ask a native English speaker to brush up English.

3.       The membrane what you studied their resistance is not economically feasible and is difficult to use at large scale. So, what is your suggestion and recommendation on this?  

4.       Using Carbon paper platinum plating catalyst on cathode electrode has two disadvantage: i) it is expensive due to platinum and ii) due to its reactivity, it might be lost its activity and affect electron transfer kinetics by forming sluggish. So, why not you focused on low-cost and other composite cathode materials?

5.       You used Carbon felt as anode? The contact surface areas of this materials is too low and decreases the growth of electroactive microorganism and this affects the performance of your cell. So, what is your recommendation?

6.       Why you select 1000 Ω load during your measurements?

7.       In the case of Catholyte, its reduction half potential must be higher and be very close to oxygen as much as possible.  Hence, by considering this most of the time dichromate, permanganate and oxygens are common. But why you choose NaCl? Not only can this chloride easily react with platinum that found on Carbon paper platinum plating catalyst on cathode. So, this are highly affects your materials performance. Then, how could you control this?

8.       From your synthetic wastewater you used acetate as a carbon source but from various literature reports using glucose as a fuel has more advantage to increase the community of microorganisms. Justify why you prefer acetate?

9.       What was the distance between anode and cathode electrode?

Reviewer 2 Report

In general, the article is tackling a very important and interesting topic, discussing the the change of microcosmic environment in forward osmosis membrane, here are some suggestions:
1) The authors should add to the introduction an explicit description of the novelty of the work and provide a comparison with the approaches available in the literature for measuring resistances in membrane systems.
2) The sentence in lines 456-457 requires clarification and reference to the source.
3) The sentence in lines 317-319 requires clarification and reference to the source
4) In this paper, AC and DC methods are used to measure film resistance. Are the test conditions the same? What are the advantages and disadvantages of the two methods
5) Usually, the film resistance test is obtained by the difference between the overall impedance and the non film impedance. Why is this method not used in this paper
Based on these issues, I think they need to be solved in minor revision.

Round 2

Reviewer 1 Report

All the comments are carefully addresed. Therefore, I recommend the paper is published in the present form in Membranes Journal. 

.